# Standardized Clinical Profiling in Spanish Patients with Chronic Tinnitus

**DOI:** 10.3390/jcm11040978

**Published:** 2022-02-13

**Authors:** Elisheba Haro-Hernandez, Patricia Perez-Carpena, Vishnu Unnikrishnan, Myra Spiliopoulou, Jose A. Lopez-Escamez

**Affiliations:** 1Department of Otorhinolaryngology, Hospital Clinico Universitario San Cecilio, 18016 Granada, Spain; elishebahh@gmail.com; 2Otology & Neurotology Group CTS 495, Department of Genomic Medicine, GENYO, Centre for Genomics and Oncological Research, Pfizer/University of Granada/Andalusian Regional Government, 18016 Granada, Spain; percarpena@gmail.com; 3Instituto de Investigacion Biosanitaria (ibs. GRANADA), Universidad de Granada, 18012 Granada, Spain; 4Sensorineural Pathology Programme, Centro de Investigación Biomédica en Red en Enfermedades Raras, CIBERER, 28029 Madrid, Spain; 5Department of Otolaryngology, Hospital Universitario Virgen de las Nieves, 18014 Granada, Spain; 6Knowledge Management and Discovery Lab, Otto-von-Guericke University Magdeburg, 39120 Magdeburg, Germany; vishnu.unnikrishnan@ovgu.de (V.U.); myra@ovgu.de (M.S.); 7Division of Otolaryngology, Department of Surgery, University of Granada, 18016 Granada, Spain

**Keywords:** tinnitus, k-clustering, machine learning, ESIT screening questionnaire

## Abstract

Background: Tinnitus is a heterogeneous condition. The aim of this study as to compare the online and hospital responses to the Spanish version of European School for Interdisciplinary Tinnitus Research screening-questionnaire (ESIT-SQ) in tinnitus individuals by an unsupervised age clustering. Methods: A cross-sectional study was performed including 434 white Spanish patients with chronic tinnitus to assess the demographic and clinical profile through the ESIT-SQ, with 204 outpatients and 230 individuals from an online survey; a K-means clustering algorithm was used to classify both responses according to age. Results: Online survey showed a high proportion of Meniere’s disease (MD) patients compared to both the general population and the outpatient cohort. The responses showed statistically significant differences between groups regarding education level, tinnitus-related hearing disorders (MD, hyperacusis), sleep difficulties, dyslipidemia, and other tinnitus characteristics, including duration, type of onset, the report of mitigating factors and the use of treatments. However, these differences were partially confirmed after adjusting for age. Conclusions: Self-reported tinnitus surveys are a low confidence source for tinnitus phenotyping. Additional clinical evaluation is needed for tinnitus research to reach the diagnosis. Age-based cluster analysis might help to better define clinical profiles and to compare responses in ESIT-SQ among subgroups of patients with tinnitus.

## 1. Introduction

Tinnitus is the perception of subjective sound in the absence of external sound or any other sounds produced by the body itself. It is a prevalent condition that affects millions of patients in Europe [1]. Around one third of all adults report experiencing tinnitus at some time in their lives, and more than 10% have prolonged tinnitus requiring medical evaluation [2].

Although this is a common symptom, it can show different psychoacoustic characteristics, and it can be described as a ringing, beeping, buzzing, hissing, or whistling sound. In addition, it can present a different frequency or intensity, and can be perceived as a pure tone, as a narrow or broadband noise, or as consisting of different sounds (which could be considered a complex tinnitus) [3].

According to the duration of the condition, tinnitus is considered chronic when it lasts for more than three months [4,5], although several studies only consider tinnitus as a chronic condition after six months of duration [2,6]. Tinnitus can occur together with other diseases, such as high-frequency hearing loss, hyperacusis, anxiety, depression, high blood pressure, Meniere’s disease (MD), vestibular schwannoma, intracranial hypertension, and migraines [7]. In fact, conditions such as anxiety or depression are considered as co-morbidities that contribute to the development of tinnitus disorder [8].

The perceived impact of tinnitus can present a large interpersonal variability, and it can also vary over time. In addition, tinnitus can be considered as an annoying symptom in a small subgroup of patients [9]. There are no objective tests to determine the existence or severity of tinnitus, but there are many different standardized questionnaires to assess quality of life and annoyance related to tinnitus. Some of the most popular questionnaires are the tinnitus handicap inventory (THI) [10,11], the tinnitus reaction questionnaire (TRQ) [12], the tinnitus questionnaire (TQ), the tinnitus functional Index (TFI) [13], and the mini-TQ [14].

In recent years, online surveys have increased and gained more importance, with the objective of creating large databases for epidemiological and clinical studies; however, the value of online surveys to predict responses in patients with tinnitus has seldom been investigated. Also, the COVID-19 pandemic has promoted teleconsultation, and the interaction with patients through online tools [15].

Almost all studies based on online surveys share as main limitations the reliability of responses and the difficulty to extent the results obtained through these questionnaires to different populations [16]. In addition, although data collection from online surveys has several advantages to both participants and researchers such as accessibility, cross-sectional studies should also include demographic data to avoid a selection bias by excluding certain subgroups of the study population [17].

The application of these online questionnaires in patients with tinnitus has been also reported [18]. A new tinnitus questionnaire was designed to collect clinical and demographic data from the population through standardized measures [19]. This questionnaire was developed by the European School for Interdisciplinary Tinnitus Research (ESIT), and it emerges as a powerful tool for a better understanding tinnitus heterogeneity and to create a tinnitus profiling framework [20]. The ESIT screening questionnaire (ESIT-SQ) is a self-reported questionnaire for standardized collection of sociodemographic and clinical information from both tinnitus and non-tinnitus individuals. It was developed with specific attention to the breadth of questions about potential risk factors and tinnitus characterization, and it has been translated into six languages, including Spanish.

The aim of this study is to perform a comparative analysis of the results obtained through the Spanish version of ESIT-SQ between two different groups of individuals, one recruited in the otorhinolaryngology clinics at two tertiary hospitals in Granada, Spain and another cohort obtained with the online version of the ESIT-SQ.

## 2. Materials and Methods

### 2.1. Study Design and Participants

A cross-sectional study was performed including a total of 434 patients with chronic tinnitus to assess the demographic and clinical profile by using the ESIT-SQ. The first group of individuals consisted of 204 outpatients visited at Departments of Otolaryngology at ‘Hospital Universitario Clínico San Cecilio’ and ‘Hospital Universitario Virgen de las Nieves’ from Granada (Spain). The second group included 230 individuals who were recruited through an online survey that was promoted in the hospital website and through social media.

The inclusion criteria for all participants were: adult individuals (>18 years old) who had been suffering from tinnitus for at least six months, regardless of the hearing loss threshold. Exclusion criteria were restricted to major diseases that could influence the responses in the ESIT-SQ (i.e., presence of an acute psychotic illness or addiction disorder, acute otological disease, such as an acute otitis, chronic otitis, a vertigo crisis, or any other ear condition apart from tinnitus itself). Individuals unable to understand Spanish to complete ESIT-SQ and records with incomplete data were also excluded. Due to the demographic characteristics of our sample, the included population consisted primarily of the white Spanish population, although ethnicity was not considered an exclusion criterion itself.

The ESIT-SQ consists of 39 closed, mainly multiple-choice, questions structured in two parts [20]. Part A includes 17 questions that can be answered by everyone irrespective of whether or not they have tinnitus. The last of these questions screens for presence of tinnitus lasting for more than five minutes over the past year. Participants that respond ‘yes’ to this question are prompted to answer 22 more tinnitus-relates questions in Part B. The patients were evaluated by an ENT specialist, and they completed the questionnaire as part of the routine clinical practice.

This questionnaire was designed by a multidisciplinary panel of epidemiologists, psychologists and otolaryngologists from ESIT to obtain a comprehensive assessment of tinnitus, other comorbid conditions, quality of life, sociodemographic data, and tinnitus characteristics [20].

### 2.2. Main Variable Description

The ESIT-SQ includes three groups of variables: sociodemographic data, comorbid conditions and tinnitus characteristics. Sociodemographic data included age, sex, height, weight, studies level, smoking habits, and familiar’s member with tinnitus and dizziness.

We categorized some variables such as hyperacusis, the number of relatives with dizziness or tinnitus, and the WHO classification for the body mass index (underweight, normal weight, overweight and obesity). Other comorbid conditions included frequency of dizziness, otological disease, surgical history, hearing loss, hyperacusis, audiological devices, pain (including headache and earache), and medical history (neurological disorder, sleep-disorder and psychiatric disorder specially). Hyperacusis was also categorized into two groups according to whether it was considered severe or not by the respondent.

The tinnitus characterization included the tinnitus frequency, duration (permanent or occasional), time since the onset of the tinnitus, duration of the disorder, number of different sounds, identifiable triggers, intensity fluctuation, rhythm, mitigating or aggravating circumstances, previous medical care, and previous treatments.

### 2.3. Statistical Analysis

First, we performed an exploratory statistical analysis for each group of participants (outpatients and online groups), separately. Since both groups had significant differences in sex and age distribution, we adjusted these variables to compare both groups in a second set of analyses. Since the online survey was distributed among members of the Spanish Association of patients with MD (“Asociación Síndrome de Meniere España”, ASMES), a large proportion of respondents reported a diagnosis of MD. On account of this bias, a stratified analysis was also performed to compare tinnitus profile in patients with or without MD.

The corresponding ordinal variables were handled as continuous variable in the analysis. Categorical variables were coded as dummy variables. For data analysis, a total of 205 variables from the baseline measurements were used.

Differences between groups were analyzed by contingency tables using *t*-tests and χ^2^-tests for independent samples, including odds ratio with 95% confidence interval. Quantitative variables following normal distribution were expressed in mean +/− standard deviations (SD). On the other hand, variables not following normal distribution were summarized through medians and interquartile ranks (25–75%). Qualitative variables were summarized through absolute and relative frequencies. The *p*-values were adjusted for multiple testing using the Bonferroni correction. A significance level of *p* ≤ 0.05 was used for all statistical tests. All statistical analyses were performed using the software package SPSS v25.0 (IBM Corporation, Armonk, NY, USA).

### 2.4. Age Clustering

Clustering is an “unsupervised” machine learning technique that aims to find groups in a dataset [21]. This technique accepts as input the number of clusters or groups in the data that the human expert expects to find, which the algorithm then discovers in a way that members of one cluster are maximally similar. The clustering technique used in this work was a “K-Means” cluster algorithm. ‘K’ points were randomly selected as the centers of each of the ‘k’ clusters, and the rest of the data points were iterated through, assigning each point to the cluster center nearest to it (for e.g., a 45-year-old patient is closer to a cluster centered at age 25 than it is to a cluster centered at age 75). After each data point was assigned to the closest cluster, the algorithm updates the cluster centers themselves as the average of all of the points that were assigned to that cluster. Intuitively, this means that the center was updated to be the ‘middle’ of all of the points it contains. This change in the cluster center may now bring points that were previously closer to other clusters to change their allegiance, causing further changes to the cluster centers in the next step, etc. This process was repeated until the assignment of the data to the clusters ‘stabilizes’. In this work, clustering was only used as a data-driven way to partition the patients into young and old groups, within each of which the analysis of variables that differed between the online and outpatient cohorts may be investigated among people of similar ages.

In our approach, responses from outpatients and the online survey were merged and clustered into two groups (young and old individuals) following the k-means algorithm. Online and outpatient responses were compared for each variable in young and old individuals. A significant *p*-value for an odd ratio (OR) < 1 means that outpatients more frequently reported this variable. The significance threshold for the *p*-value after Bonferroni correction was <0.00263.

## 3. Results

### 3.1. Description of Both Samples

On the first part of the analysis, we describe raw data in both samples. A total of 434 patients with tinnitus were recruited; 204 (46.3%) were enlisted in the outpatient clinic at the Department of Otorhinolaryngology and 230 (53.7%) by online media.

There were significant differences in the distribution of age, sex, and level of education (*p*-value < 0.001) between both surveys. The average age in the outpatient cohort was 55 (46–62) years old, while in online group it was 44 (37–53) years old. Female patients were 66% and 52% in the online and outpatients survey, respectively. Several differences were also observed among the clinical features both surveys. The reported frequencies of acoustic trauma, acute otitis media, and presbycusis they did not reach statistically significant differences (*p* > 0.05). On the other hand, sleep disorder (*p* < 0.001) and metabolic illnesses such as dyslipidemia (*p* < 0.001) were higher in the outpatient cohort. Of note, the reported frequency of Meniere’s disease (*p* < 0.001), vertigo (*p* < 0.001), and hyperacusis (*p* < 0.001) was different between both samples, with higher rates in the online survey (*p* < 0.001). Anxiety and depression were not different between both groups. Table 1 summarizes the comparisons in terms of demographic characteristics and comorbid conditions between both groups.

The duration of tinnitus showed significant differences 72 (36–132) months in the online survey and 24 (12–96) months in the outpatient survey (Table 2). Moreover, the time since tinnitus became a bothersome symptom was 48 (24–120) months and 24 (12–72) months in the online and outpatient surveys, respectively. Regarding the characteristics of perceived tinnitus, the number of sounds reported in the online survey was not statistically significant (*p* > 0.05). However, the tinnitus onset was different (*p* < 0.001), with a higher tendency to develop sudden onset in the online compared to the outpatient survey. According to influencing factors on tinnitus, there was a higher number of individuals reporting both aggravating and mitigating factors on tinnitus in the online group compared to the outpatient survey, although these differences were only statistically significant for the mitigating factors (*p* = 0.001) that was higher in the online group. Tinnitus increasing factors includes situations or conditions that could potentially worsen the tinnitus, such as lack of sleep, stress or alcohol/coffee consumption. On the other hand, tinnitus-reducing factors could decrease the intensity of tinnitus or improve its perception.

In addition, the number of treatments used for tinnitus was analyzed, recording a higher proportion in the online survey (*p* < 0.001), but there were no differences between all of the subgroups of treatment.

### 3.2. Age and Sex Adjusted Comparison of Both Surveys

In the second part of analysis, both samples were adjusted by age and sex to compare the clinical and psychoacoustic variables. Finally, we retrieved a sample of 344 individuals (204 outpatients and 140 respondents to the online survey). In this case, the average age was 52 (46–57.75) years old in the online group and 55 (46–62) years old in the outpatient cohort. Women represented 62% of the on-line group and 52.2% of the outpatient cohort.

The level of education was significantly higher in online respondents than in outpatients (*p* < 0.001). The clinical profile was different in outpatients and online participants. MD (75%, *p* < 0.001), vertigo (*p* < 0.001) and hyperacusis (88%, *p* < 0.001) were more commonly reported in the online survey. However, hyperacusis was also frequently reported in outpatients (69%). Table 3 compares demographic data and comorbid conditions between both groups after adjusting for sex and age.

There were differences in the time since the onset of tinnitus 96 (36–180) months in the online survey and 24 (12–93) months in outpatient group, respectively (*p* < 0.001, Table 4). These differences were also for the time since the tinnitus began being a discomfort 60 (24–129) months and 24 (12–60) months in online and outpatients survey, respectively *p* = 0.004). There were also differences in the number of perceived sounds in both groups. Most of the outpatients reported that their tinnitus consisted of one sound, while most of the patients from the online survey reports two or more sounds (*p* < 0.001). No trigger associated with tinnitus was significantly different between both groups, the only exception being a change in hearing (*p* = 0.002), which was more frequently reported in the online survey. Aggravating factors were also more reported by the online group than by outpatients (*p* < 0.001).

Some variables, which were statistically significant in the whole sample, were not statistically significant after age and sex adjustment (sleep disorders and dyslipidemia between the comorbid conditions). On the other hand, the tinnitus characteristics that were significantly different between both samples were the type of onset, mitigating factors and treatment necessity. However, when both groups were adjusted for age and sex, the differences were the duration of disturbing tinnitus, the number of sounds, changes in hearing, and aggravating factors.

#### Clustering Approach for Adjusting the Sample on Age

For the K-means clustering, the number of clusters k was set to 2, since this yielded clusters that were balanced in size, without outliers, and also resulted in clusters large enough to support most of the statistical comparisons performed. The ‘young’ cluster included 160 individuals, aged between 19 and 53, with a mean age of 44.77 ± 7.33, and a median age of 46. This group had 80 online participants and 79 outpatients visited at the hospital. One hundred fifty-two patients aged between 54 and 94 belonged to the ‘old’ cluster, with a mean age of 62.04 ± 6.62 and a median age of 61. This group had 52 online respondents and 98 outpatients. The fact that the mean and the median ages in each of the clusters is comparable shows that the process was not influenced by outliers.

Statistical comparisons between the online and outpatient surveys were performed using the chi squared test for each of the variables and each cluster. The *p*-values for significance were adjusted for multiple comparisons using the Bonferroni correction, yielding the adjusted *p*-values for 0.1, 0.05 and 0.01 as 0.0053, 0.00263, and 0.0005, respectively. At the adjusted significance level of 0.00263, online and outpatient surveys in both the young and the old clusters were significantly different in the frequency of ear problems reported. The other variables that were different between the two groups were the occurrence of MD (*p* = 5.18 × 10^−12^ for young, and *p* = 1.39 × 10^−13^ for old cluster), which showed a significant difference between the online and outpatient cohorts in both clusters of young and old patients and was more likely to be reported among outpatients. On the other hand, hyperacusis showed a significant difference between online and outpatient cohorts only in the cluster of young patients, whereas the online group reported higher rates of this condition hyperacusis. However, after analyzing the intensity of hyperacusis, we found that the older respondents were more likely to report severe hyperacusis in the online survey (*p* = 0.001) (Table 5).

### 3.3. Stratified Analysis for Meniere’s Disease

We merged all online and outpatient surveys and classified them according to the presence of MD. Each variable was compared using *t*-tests or χ^2^-tests, accordingly. These analyses showed statistically significant differences between both groups in terms of age and sex, the history of acute otitis media, vertigo, and hyperacusis (Table 6).

The tinnitus profile was different in individuals with MD compared with non-MD participants (Table 7). Tinnitus duration, the duration of debilitating tinnitus, the worry on tinnitus, and the number of sounds perceived was higher in patients with MD. Particularly, changes in hearing (OR = 3.39, *p* < 0.001) and the description of tinnitus increasing factors for tinnitus (OR = 3.95, *p* < 0.001) were reported most frequently in MD.

## 4. Discussion

The purpose of this study was to investigate the clinical profile of Spanish patients with chronic tinnitus by using the ESIT-SQ in two different groups of individuals, namely outpatients and online participants. Although the ESIT-SQ is a very detailed self-reported questionnaire, it records information on many diseases that should be supported by the diagnosis of a clinician, such as MD, hearing loss, or anxiety/depression. According to this, the ESIT-SQ should be used as a screening instrument and any disease or disorder must be confirmed by a clinical diagnosis. Despite these limitations, our study was able to identify a set of characteristics that are often present in participants with tinnitus.

Although the ESIT-SQ was developed by Genitsaridi et al., only few studies have used it [20]. There is also a lack of studies comparing differences in population characteristics between face-to-face and self-report questionnaires. However, the use of online questionnaires to study tinnitus has already been reported [22]. This study recorded the presence of some of the physical symptoms included in the ESIT-SQ, such as neck pain and headache through an online questionnaire. The presence of neck pain was higher in this study compared to our outpatient and online cohorts, but headache was reported in fewer patients as compared to our sample [22,23].

Most individuals from the online survey were patients with chronic tinnitus and MD, a chronic inner ear disorder defined by episodes of vertigo associated with tinnitus and sensorineural hearing loss [24,25]. Although this could be considered a bias for the online survey, these results may contribute to a better understand of tinnitus profile in MD. This study had the limitations of all cross-sectionals’ designs. Therefore, we could not rule out the possibility of reverse causality. Regardless, the analysis of two samples of individuals with tinnitus from the same population can be useful to find differences in anonymous responses obtained online with the responses given by patients with confirmed chronic tinnitus at the hospital.

Regarding the characteristics of each group, there are few studies that have investigated the effect of age on the responses obtained to assess comorbidities in patients with tinnitus. There are some studies where the average age was higher [26]. However, this study analyzed aging population, and included patients over 45 years old. Our results showed differences between both groups regarding the education level. Both groups showed a high educational level. However, the level of education was significantly higher in the online population. One study showed a lower educational level in patients with tinnitus compared to healthy controls, although the results were not statistically significant [26]. Another study showed an association between a lower level of education and tinnitus. This association was not confirmed after a multiple logistic regression analysis [27]. We could partially explain our findings through the characteristics of the online sample, consisting of individuals with higher knowledge in technology which was not necessary in the outpatient sample.

The ESIT-SQ recorded information about otological disorders such as acoustic trauma, infections, or different types of hearing loss. Hearing loss has shown a significant association with tinnitus in several studies [28,29]. However, other study reported that hearing loss has low impact on perceived tinnitus [23]. Their results showed differences between tinnitus intensity and its minimum masking level, which could be explained by cochlear damage. In this regard, the ESIT-SQ could not distinguish the type of hearing loss, its laterality, nor the hearing threshold.

Vertigo was more frequently reported in online survey. This could be explained by the higher prevalence of MD in this cohort. However, dizziness and vertigo are non-specific symptoms that could be explained by multiple causes including vestibular, brain, or heart disorders [30,31,32,33]. Headache, sleep disorders, or stress could be related with both symptoms [34].

Stress [35,36], depression, and anxiety [35,37,38] have been widely related to tinnitus. In our study, one third of the individuals reported anxiety in the online survey and a quarter in the outpatient cohort. A recent case-control study reported that patients with severe tinnitus suffered from more psychiatric disorders, with higher prevalence of depression and somatization [39]. Sleep-disorder was a frequent complaint in our samples, and it showed a statistically significant difference between both groups, with higher rates in outpatients than in online participants. However, this difference was missed after adjusting by sex and age. In addition, it has been described that females with tinnitus are more frequently affected by sleep disorders [40].

We did not observe a higher prevalence of temporomandibular joint disorders, in contrast with other prevalence studies [41,42]. Hypertension, dyslipidemia, diabetes mellitus, stroke, angina, and myocardial infarction have been proposed as potential risk factors for tinnitus [28,29], with controversial results [29,43]. In our comparison of both samples, dyslipidemia was more frequently reported among outpatients than in the online cohort. However, after adjusting by sex and age, the rates of hypertension and dyslipidemia were not different between both samples. In addition, our study did not find differences between both groups in terms of body mass index (BMI), and about 20% of both samples had obesity. One study reported that obesity may decrease the susceptibility for tinnitus [43], while other one reported a higher risk for tinnitus in individuals with BMI > 30kg/m^2^ [28,44]. Nonetheless, a BMI ≥ 30 kg/m^2^ could be related to higher rates of dyslipidemia and hypertension in tinnitus patients.

Finally, we also observed differences on hyperacusis between both cohorts, and it affected a significant percentage of patients with tinnitus. Hyperacusis was more frequent in the online survey. This could be explained due to a significantly higher proportion of patients of MD in this group. Other studies have also described a higher frequency of hyperacusis in patients with tinnitus than individuals without tinnitus. It is well known that tinnitus is particularly common in patients with hyperacusis, and they can co-occur in patients with severe tinnitus [35,45]. In addition, both conditions are widely associated and could share a common pathophysiology [46].

There are multiple variables that may explain tinnitus heterogeneity, but there is a lack of studies focused on the characteristics of tinnitus. Genitsaridi et al. summarized in their systematic review what symptoms were more frequently recorded in the studies and which showed significant differences among different groups of patients. Among all of these characteristics, the most frequently reported were tinnitus severity, hearing ability, age, and depressive symptoms [38].

Regarding duration of tinnitus, we find significantly differences between both groups, with a higher duration in online survey. Other studies showed similar results where the duration of tinnitus was reported as an important variable (although these results were not compared to a control group) [38].

When we analyzed the differences between patients with MD, we observed that changes in hearing and triggers are associated with tinnitus in MD. Moreover, hyperacusis and worry about tinnitus were more prevalent in these patients. These findings were already reported in MD [47]. In addition, it has been described a higher prevalence of anxiety, measured through questionnaires such as the hospital anxiety and depression scale (HADS). This increase in anxiety could be explained by the fact that our patients with MD were recruited by volunteer questionnaire, and since these patients are more aware about their disease due to its severity. In addition, it has been described that HADS scores showed a positive correlation with another quality-of-life questionnaires, such as the THI or the visual analogue scale on tinnitus annoyance [3].

There are controversies in the effect of different treatments on tinnitus, as their effectiveness is limited. Some studies reported the effect of several treatments in tinnitus, although there are limitations due to the heterogeneity of the results [48]. Other study reported gender differences in the effect of treatment in terms of tinnitus-related distress and depression severity [40]. Similar gender-related differences in the effectiveness of the treatment were also reported in other study [38]. Although our results were not stratified by sex, we observed differences between outpatients and online cohorts.

There is a lack of literature on psychoacoustic characterization of tinnitus and there are no studies that compared two populations with tinnitus with different recorded way. New investigations with electro-acoustical devices are needed to standardize psychoacoustic measurements and to design and validate new tools for sound therapy to treat them.

By using age clustering to improve clinical profiling, the separation between young and old patients confirmed some findings reported in the non-stratified comparative method. On the other hand, some other significant differences, such as level of education, presbycusis, and dyslipidemia were no longer replicated after age clustering. In addition, after Bonferroni correction, the differences in acoustic trauma in young patients from the online cohort and outpatients remained, while differences in hyperacusis and neck pain in old patients were not significant. The variables that were found to be significantly different between online participants and outpatients were ear problems, MD, hyperacusis for the younger group, and severe hyperacusis for the older cluster.

## 5. Conclusions

Self-report questionnaires such as ESIT-SQ are useful to standardize clinical profiling in individuals with chronic tinnitus. However, these instruments should be used as screening and all medical or psychological conditions such as anxiety and depression, hearing loss, or MD, should be supported by a medical diagnosis.

There are differences in the tinnitus profile in the online and outpatient surveys including the level education, hearing disorders associated with tinnitus (MD and hyperacusis), and difficulty to sleep. Moreover, tinnitus characteristics were different between samples, including the type of onset, tinnitus duration, the report of mitigating factors, and the use of treatments. When both groups were adjusted for age and sex, there were still differences in the tinnitus profile including the total duration of tinnitus, duration of debilitating tinnitus, number of perceived sounds, subjective changes in hearing, and factor increasing tinnitus. Age clustering indicates the role of acoustic trauma in young patients with tinnitus.

## Figures and Tables

**Table 1 jcm-11-00978-t001:** Demographic data and comorbid conditions for online sample and outpatients with chronic tinnitus (*n* = 434).

Variable	Category	Online Sample (*n* = 230)	Outpatients (*n* = 204)	Corrected *p*
**Age (years)**		44 (37–53)	55 (46–62)	<0.001
**Sex**				<0.001
	Female Male	151 (66%) 79 (34%)	105 (52.20%) 99 (47.80%)	
Body mass Index				>0.050
	Underweight	6 (2.60%)	7 (3.70%)	
	Normal-weight	110 (47.80%)	79 (41.40%)	
	Overweight	72 (31.30%)	70 (36.60%)	
	Obesity	42 (18.30%)	35 (18.30%)	
**Level of education**				<0.001
	No school	0 (0%)	3 (1.50%)	
	Primary school	19 (8.30%)	48 (23.50%)	
	Middle school	28 (12.20%)	50 (24.50%)	
	High school	44 (19.10%)	41 (20.10%)	
	University/higher degree	139 (60.40%)	60 (29.40%)	
Tinnitus family history				>0.050
	No	73 (52.10%)	115 (56.40%)	
	Three or less relatives	53 (37.90%)	79 (38.70%)	
	More than three relatives	14 (10%)	10 (4.90%)	
Ear condition	Acoustic trauma	12 (5.20%)	21 (10.30%)	>0.050
	Acute otitis media	7 (3.10%)	19 (9.30%)	>0.050
	Presbycusis	3 (1.30%)	9 (4.40%)	>0.050
	SSHL *	22 (9.60%)	16 (7.80%)	>0.050
	**Meniere’s disease**	164 (71.60%)	32 (15.70%)	<0.001
**Hyperacusis**		198 (86.1%)	141 (69.10%)	<0.001
**Vertigo**				<0.001
	Never	30 (13.1%)	62 (30.50%)	
	One or less per year	40 (17.50%)	23 (11.30%)	
	More than two per year	159 (69.40%)	107 (52.70%)	
	No answer	0	11 (5.40%)	
Pain	Headache	122 (53%)	106 (52%)	>0.050
	Neck pain	88 (38.30%)	104 (61.30%)	>0.050
	Ear pain	60 (26.10%)	37 (18.10%)	>0.050
Psychiatric conditions	Anxiety	77 (33.50%)	51 (25%)	>0.050
	Depression	39 (17%)	27 (13.20%)	>0.050
**Sleep-disorders**	Start	49 (21.30%)	74 (36.30%)	<0.001
Heart conditions	HBP **	22 (9.60%)	34 (16.70%)	>0.050
**Metabolic disorders**	**Dyslipidemia**	27 (11.7%)	52 (25.5%)	<0.001

* SSHL, sudden sensorineural hearing loss; ** HBP, high blood pressure.

**Table 2 jcm-11-00978-t002:** Tinnitus characteristics in patients with chronic tinnitus (*n* = 434).

Variable	Category	Online Sample *n* = 230	Outpatients *n* = 204	Corrected *p*
**Tinnitus duration (months)**		104.53 ± 95.2	73.70 ± 99.80	0.036
**Debilitating** tinnitus duration (months)		84.88 ± 82.44	63.88 ± 94.14	>0.050
Worry on tinnitus				>0.050
	Severely	93 (41.30%)	81 (40.50%)	
	Moderately	90 (40%)	76 (38%)	
	Slightly	28 (12.40%)	19 (9.50%)	
	Not at all	6 (2.70%)	19 (9.50%)	
	NA	8 (3.60%)	5 (2.50%)	
Number of sounds				>0.050
	One or less	79 (34.60%)	54 (27%)	
	More than one	145 (63.60%)	77 (38.50%)	
	No answer	4 (1.70%)	69 (34.50%)	
**Type of onset**				<0.001
	Sudden	84 (36.50%)	113 (56.50%)	
	Gradual	87 (37.20%)	78 (39%)	
	No answered	55 (24.30%)	9 (4.40%)	
Triggers	Changes in hearing	43 (19.40%)	19 (9.50%)	>0.050
	Ear fullness	83 (37.40%)	50 (25.10%)	>0.050
	Stress	59 (26.60%)	46 (23.10%)	>0.050
**Tinnitus increasing factors**		208 (92.40%)	135 (69%)	0.054
**Tinnitus reducing factors**		159 (71.30%)	112 (57.40%)	<0.001
**Treatment**		41.4/58.6 (%)	21.1/75.9//3 (%)	<0.001
**(Yes/No//NA)**				

NA, no answer.

**Table 3 jcm-11-00978-t003:** Demographic data and comorbid conditions in patients with chronic tinnitus after adjusting for sex and age (*n* = 344).

Variable	Category	Online Sample (*n* = 140)	Outpatients (*n* = 204)	Corrected *p*
Age (years)		52 (46–57.75)	55 (46–62)	>0.050
Sex				>0.050
	Female	86 (62%)	105 (52.20)	
	Male	53 (38%)	96 (47.80%)	
Body mass Index				>0.050
	Underweight	3 (2.10%)	4 (2.10%)	
	Normal-weight	62 (44.30%)	82 (43%)	
	Overweight	45 (32.10%)	70 (36.60%)	
	Obesity	30 (21.40%)	35 (18.30%)	
**Level of education**				<0.001
	No school	0 (0%)	3 (1.50%)	
	Primary school	16 (11.40%)	48 (23.50%)	
	Middle school	20 (14.30%)	50 (24.50%)	
	High school	31 (22.10%)	41 (20.10%)	
	University/higher degree	73 (52.10%)	60 (29.40%)	
Tinnitus family history				>0.050
	No	73 (52.10%)	115 (56.40%)	
	Three or less relatives	53 (37.90%)	79 (38.70%)	
	More than three relatives	14 (10%)	10 (4.90%)	
Ear condition	Acoustic trauma	8 (5.70%)	21 (10.30%)	>0.050
	Acute otitis media	5 (3.60%)	19 (9.30%)	>0.050
	Presbycusis	3 (2.10%)	9 (4.40%)	>0.050
	SSHL *	14 (10%)	16 (7.80%)	>0.050
	**Meniere disease**	104 (75%)	32 (15.70%)	<0.001
**Hyperacusis**		198 (88%)	141 (69.10%)	<0.001
**Vertigo**				**<0.001**
	Never	17 (12.1%)	62 (30.5%)	
	One or less per year	22 (15.7%)	23 (11.3%)	
	More than two per year	101 (72.1%)	107 (52.7%)	
	**No answer**	0	11 (5.4%)	
Pain	Headache	68 (48.6%)	106 (52%)	>0.050
	Neck pain	49 (35%)	104 (51%)	0.054
	Ear pain	35 (25%)	37 (18.10%)	>0.050
Psychiatric conditions	Anxiety	44 (31.4%)	51 (25%)	>0.050
	Depression	21 (15%)	27 (13.20%)	>0.050
Sleep-disorders	Start	32 (23%)	74 (36.50%)	>0.050
Heart conditions	HBP **	20 (14.30%)	34 (16.70%)	>0.050
Metabolic disease	Dyslipidemia	22 (15.7%)	52 (25.50%)	0.054

* SSHL, sudden sensorineural hearing loss; ** HBP, high blood pressure. Variables with a *p* value < 0.05 have been marked in bold.

**Table 4 jcm-11-00978-t004:** Tinnitus characteristics in hospital outpatients and online survey participants after adjusting for sex and age (*n* = 344).

Variable	Category	Online Sample (*n* = 140)	Outpatients Sample (*n* = 204)	Corrected *p*
**Tinnitus duration (months)**		119.25 ± 101.37	72.84 ± 99.45	<0.001
**Debilitating tinnitus duration (months)**		95.78 ± 94.04	60.54 ± 92.96	0.036
Worry on tinnitus				>0.050
	Severely	58 (41.70%)	81 (40.50%)	
	Moderately	52 (37.40%)	76 (38%)	
	Slightly	19 (13.70%)	19 (9.50%)	
	Not at all	4 (2.90%)	19 (9.50%)	
	No answered	6 (4.30%)	5 (2.50%)	
**Number of sounds**				<0.001
	One or less	47 (34%)	113 (56.50%)	
	More than one	89 (64%)	72 (39%)	
	No answered	3 (2%)	9 (4.50%)	
Type onset				>0.050
	Sudden	49 (35.80%)	54 (27%)	
	Gradual	58 (42.30%)	77 (38.50%)	
	No answered	30 (21.90%)	69 (34.50%)	
Triggers	**Changes in hearing**	31 (23.10%)	19 (9.50%)	0.018
	Ear fullness	40 (30%)	50 (25.10%)	>0.050
	Stress	30 (22.40%)	46 (23.10%)	>0.050
**Tinnitus increasing factors**		128 (92.80%)	135 (69%)	<0.001
Tinnitus reducing factors		92 (67.60%)	112 (57.40%)	>0.050
**Treatment**		26.8/63.2 (%)	21.1/75.9//3 (%)	0.054
**(Yes/No/NA)**				

NA, no answer.

**Table 5 jcm-11-00978-t005:** Comparison of the main clinical variables in the ESIT-SQ according to the age of patients with chronic tinnitus.

Variable	Cluster	χ2	OR (95% CI)	Corrected *p*
Sex	Young	3.323	0.53 (0.28–1.00)	0.680
	Old	0.128	0.83 (0.42–1.64)	0.721
Level education	Young	7.961	0.39 (0.20–0.77)	0.047
	Old	10.448	0.37 (0.18–0.76)	0.015
**Otological disease**	Young	26.166	**0.05 (0.01–0.20)**	**<0.001**
	Old	23.828	**0.05 (0.01–0.22)**	**<0.001**
Acoustic trauma	Young	5.472	6.31 (1.35–29.47)	0.019
	Old	0.001	1.19 (0.35–4.06)	0.975
Presbyacusis	Young	0.000	0.50 (0.04–5.62)	0.991
	Old	0.900	3.85 (0.46–32.16)	0.343
Acute Otitis	Young	1.617	2.89 (0.74–11.33)	0.203
	Old	0.698	2.48 (0.51–11.92)	0.404
Neck Pain	Young	0.004	1.08 (0.57–2.04)	0.950
	Old	7.251	2.74 (1.36–5.51)	0.007
Sleep Disorder	Young	1.325	1.64 (0.79–3.41)	0.250
	Old	1.328	1.62 (0.79–3.30)	0.249
High blood pressure	Young	0.049	0.79 (0.29–2.11)	0.824
	Old	0.011	1.15 (0.49–2.66)	0.917
Low blood pressure	Young	0.000	1.37 (0.30–6.32)	0.986
	Old	0.000	1.21 (0.36–4.14)	0.997
Cholesterol	Young	0.269	1.46 (0.55–3.83)	0.604
	Old	1.645	1.21 (0.35–4.14)	0.200
**Meniere’s Disease**	**Young**	47.614	**0.08 (0.03–0.17)**	**<0.001**
	**Old**	54.719	**0.05 (0.02–0.12)**	**<0.001**
**Hyperacusis (yes/no)**	**Young**	9.198	**0.27 (0.11–0.61)**	**0.002**
	Old	4.703	0.34 (0.14–0.85)	0.030
**Severity of hyperacusis**	Young	0.029	1.17 (0.52–2.65)	0.864
	**Old**	11.518	**4.06 (1.84–8.97)**	**0.001**
Anxiety	Young	0.002	0.96 (0.48–1.91)	0.960
	Old	3.231	0.47 (0.22–0.99)	0.072
Depression	Young	0.756	0.61 (0.24–1.49)	0.385
	Old	0.051	1.25 (0.48–3.27)	0.822
Antidepressant	Young	0.000	1.00 (0.24–4.15)	1.000
	Old	0.204	1.93 (0.39–9.66)	0.651
Familial history of tinnitus	Young	1.088	0.68 (0.36–1.27)	0.297
	Old	0.017	0.99 (0.50–1.94)	0.896

Cluster analysis: The “Young” subgroup includes population between 19–53 years; the “Old” subgroup includes population between 54–94 years. Variables with a *p* value < 0.05 have been marked in bold.

**Table 6 jcm-11-00978-t006:** Demographic data and comorbid conditions in patients with chronic tinnitus after stratifying for Meniere’s disease (*n* = 344).

Variable	Category	Meniere’s Disease (*n* = 136)	Non Meniere’s Disease (*n* = 208)	Corrected *p*
**Age (years)**		52.26 ± 8.62	53.23± 12.75	<0.001
**Sex**				<0.001
	Female	85 (62.5%)	105 (50.70%)	
	Male	51 (37.5%)	98 (47.30%)	
Body mass Index				>0.050
	Underweight	1 (0.70%)	7 (2.10%)	
	Normal-weight	62 (45.60%)	81 (41.80%)	
	Overweight	43 (31.60%)	72 (37.1%)	
	Obesity	30 (22.10%)	35 (18%)	
Level of education				>0.050
	No school	0 (0%)	3 (1%)	
	Primary school	23 (16.90%)	64 (18.70%)	
	Middle school	21 (15.40%)	70 (20.40%)	
	High school	24 (17.60%)	72 (21%)	
	University/higher degree	68 (50%)	132 (38.50%)	
Tinnitus family history				>0.050
	No	74 (54.40%)	188 (54.80%)	
	Three or less relatives	52 (38.20%)	132 (38.50%)	
	More than three relatives	10 (7.4%)	23 (6.70%)	
Ear condition	Acoustic trauma	5 (3.70%)	24 (11.60%)	>0.050
	**Acute otitis media**	2 (1.50%)	22 (10.60%)	0.018
	Presbycusis	4 (3%)	8 (4%)	>0.050
	SSHL *	11 (8.10%)	19 (9.20%)	>0.050
**Hyperacusis**		122 (90%)	141 (68.10%)	<0.001
**Vertigo**				<0.001
	Never	2 (1.5%)	77 (37.2%)	
	One or less per year	16 (11.9%)	29 (14%)	
	More than two per year	116 (85.9%)	91 (44%)	
	No answer	1 (0.7%)	10 (4.8%)	
Pain	Headache	75 (55.1%)	98 (47.3%)	>0.050
	Neck pain	50 (36.80%)	103 (50%)	>0.050
	Ear pain	31 (22.80%)	41 (20%)	>0.050
Psychiatric conditions	Anxiety	39 (28.70%)	56 (27.10%)	>0.050
	Depression	22 (16.2%)	26 (12.60%)	>0.050
Sleep-disorders	Start	33 (24.30%)	72 (35%)	>0.050
Heart conditions	HBP **	18 (13.20%)	36 (17.40%)	>0.050
Metabolic disorders	Dyslipidemia	23 (17%)	51 (25%)	>0.050

* SSHL, sudden sensorineural hearing loss; ** HBP, high blood pressure.

**Table 7 jcm-11-00978-t007:** Tinnitus characteristics in patients with chronic tinnitus, after stratifying for Meniere’s disease (*n* = 344).

Variable	Category	Meniere’s Disease *n* = 136	Non Meniere’s Disease *n* = 208	Corrected *p*
**Tinnitus duration (months)**		125.84 ± 108	55.29 ± 87.47	<0.001
**Debilitating tinnitus duration (months)**		106.55 ± 99.60	44.04 ± 78.25	<0.001
**Worry on tinnitus**				0.018
	Severely	69 (50.70%)	70 (34.70%)	
	Moderately	49 (36%)	78 (38.6%)	
	Slightly	12 (8.80%)	26 (12.90%)	
	Not at all	1 (0.70%)	22 (10.90%)	
	No answered	5 (3.70%)	6 (3%)	
**Number of sounds**				<0.001
	One or less	44 (32.40%)	115 (56.9%)	
	More than one	89 (65.40%)	78 (38.60%)	
	No answered	3 (2.20%)	9 (4.50%)	
Type of onset				>0.050
	Sudden	43 (31.60%)	60 (30%)	
	Gradual	64 (47.10%)	70 (35%)	
	No answered	29 (21.30%)	70 (35%)	
Triggers	**Changes in hearing**	32 (24.10%)	17 (8.50%)	<0.001
	Ear fullness	37 (27.80%)	52 (26.10%)	>0.050
	Stress	30 (22.60%)	45 (22.60%)	>0.050
**Tinnitus increasing factors**		123 (90.40%)	139 (70.6%)	<0.001
Tinnitus reducing factors		89 (66%)	114 (59%)	>0.050
**Use of any treatment for tinnitus**		52 (39%)	42 (20%)	<0.001

## Data Availability

All data obtained in this study have been anonymized and are available from the corresponding author upon reasonable request.

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
