# Peer review of "Standardized Clinical Profiling in Spanish Patients with Chronic Tinnitus"

_jcm, 2022, doi:10.3390/jcm11040978_

Round 1

Reviewer 1 Report

Age clustering may improve clinical profiling in Spanish pa-2 tients with chronic tinnitus

I mostly liked this manuscript, and I think it deserves to be published. I have only uncomplicated issues that I would like to see addressed first. I would highly recommend that the authors have the paper re-read for language, as it was sometimes hard to parse meaning. I have listed a few of the language issues below, but the manuscript really needs a read-over. 

Issues to be addressed:
Methodological:

This paper uses 2 groups of patients: a group recruited online and a set of patients from a clinical population. 
- 75% of the population recruited online reports having Meniere's disease. This is really not representative of the general situation. The American Tinnitus Association for instance states that 3% of its (tinnitus) members have reported a diagnosis of Meniere. This exceedingly strong skew needs to be mentioned much earlier, possibly even in the abstract, and its origin accounted for. Otherwise it really casts doubts on the validity/usefulness of the results. 

More importantly, 
- The study uses K-means for clustering, which presents limitations largely unaddressed in the text. 
The "K" has to be chosen by hand. The chosen value of K=2 is hardly justified on a sound statistical basis in the manuscript. Almost any data can be divided in 2!

Also, for K-means to make sense, there is an assumption that the clusters will be "spherical" in the space of coordinates chosen. This should be verified. 
https://www.inovex.de/de/blog/disadvantages-of-k-means-clustering/
https://developers.google.com/machine-learning/clustering/algorithm/advantages-disadvantages

https://www.quora.com/What-are-the-drawbacks-of-K-means-clustering-1

These issues are standard in clustering, so I won't go into them here. 

-Thoroughout the paper, there is a complete lack of distinction between tinnitus perception and tinnitus reaction. This is addressed indirectly in the questionnaire(s), but is hardly distinguished in the analysis, or in the discussion (except in line 333). Some statements are ambiguous, such as (line 332) "Stress [29,30], depression and anxiety [29,31,32] have been widely related to tinnitus". Which way do you mean the causation to occur? Does it affect your results?
- Statistics are probably not correctly used at times. If your duration of symptoms is "63.88 ± 94.14 months", it is a safe bet on my part that you are not using the right stats. For this (in multiple places), you might want to use median and interquartile range, it would be more informative to the reader too. 

Comments:
- The introduction was interesting and well written, but most of it was fairly irrelevant to the paper. If there is a need to reduce paper length, this would be a good place.
- What is the meaning of the exclusion "active otological illness". Since they have tinnitus, they seem to have an active illness. Needs precision. 
- The requirement of being "caucasian" is a bit weird without justification. First of all, "White" should be used instead of "caucasian" I think, though I don't want to open that can of worms. More importantly, it limits the validity of the results to Whites only. This should be justified. It should be mentioned in the abstract.  
- The family history repeatedly refers to "three or less families" and I don't know what that means. Family members?
- How the groups of patients are referred to changes many times in the text. In just 3 paragraphs (around lines 240 to 280), the same (?) participants are referred to as "Outpatients", then as "offline participants" in the next paragraph, then as "hospital participants" in the next paragraph, then as outpatients again. Stick to one nomenclature throughout!
- Tinnitus is referred to as "disabling" multiple times, but I think you mean "debilitating". 
- Line 321: "any otological diseases could eventually lead to tympanic membrane abnormalities". That seems to be a bit of an overstatement?

Minor issues:
- line 117: why not include vertigo with dizziness?
- dyslipidemia (8 times) is spelled dyslipidaemia 3 times. Chose one!

The paper has to be read by a native speaker. Here are a few of the points that made me stumble when reading. I don't normally lists stuff like this, but I would like to see this study published and be useful. 
Line 30: However, these differences were partially confirmed
58  quality of life ant the annoyance (-> and)
58  the most extended questionnaires  (extensive?)
96  a specific call at social media
99  interfere the responses 
100 addition disorder
134 high frecuency
135 prolfile
209 there were not differences
Table 2: worry on tinnitus: what does that mean?
Table 2 and other tables: "No answered" (-> no answer)
Table 2 and other tables: "Tinnitus increasing factors": what does that mean? Co-factors that will make an episode worse? I don't know. 
Table 2: the line that states "Treatment" reports a value of 21.1/75.9 and again, I have no clue of the meaning of this. 
296  others resulted inconsistent
375  it has been described a higher prevalence of anxiety
(then in the next line "there has been described")
393 there are not studies

Author Response

I mostly liked this manuscript, and I think it deserves to be published. I have only uncomplicated issues that I would like to see addressed first. I would highly recommend that the authors have the paper re-read for language, as it was sometimes hard to parse meaning. I have listed a few of the language issues below, but the manuscript really needs a read-over.

Issues to be addressed:

Methodological:

This paper uses 2 groups of patients: a group recruited online and a set of patients from a clinical population.

- 75% of the population recruited online reports having Meniere's disease. This is really not representative of the general situation. The American Tinnitus Association for instance states that 3% of its (tinnitus) members have reported a diagnosis of Meniere. This exceedingly strong skew needs to be mentioned much earlier, possibly even in the abstract, and its origin accounted for. Otherwise it really casts doubts on the validity/usefulness of the results.

Response: Thank you for this valuable comment. We are aware of this, and have included a comment on that in the abstract, the methods section and the discussion.

More importantly,

- The study uses K-means for clustering, which presents limitations largely unaddressed in the text.

The "K" has to be chosen by hand. The chosen value of K=2 is hardly justified on a sound statistical basis in the manuscript. Almost any data can be divided in 2!

Also, for K-means to make sense, there is an assumption that the clusters will be "spherical" in the space of coordinates chosen. This should be verified.

https://www.inovex.de/de/blog/disadvantages-of-k-means-clustering/

https://developers.google.com/machine-learning/clustering/algorithm/advantages-disadvantages

https://www.quora.com/What-are-the-drawbacks-of-K-means-clustering-1

These issues are standard in clustering, so I won't go into them here.

Response: Thank you for your comment, and your willingness to explore the method a bit deeper. It is indeed true that a basic algorithm like K-Means has many disadvantages. However, since in our (special) case we applied it only on one variable, it does serve only as a data-driven segmenter into comparable age groups (ie, since the data can be divided into 2 chunks in many ways, why not discover the split point using data?). While the k=2 does feel a bit arbitrary, the choice of k in this case was done because increasing the 'k’ would leave us with too little data for making statistical comparisons within the clusters... Coupled with the fact that the means and the medians were comparable (which is the closest we could get to checking if the clusters are 'spherical' in one dimension), and that the two resultant clusters were comparable in size, k=2 does seem a reasonably good way to segment. We hope that the text that has been added in both section 2.4 and in section 3.2.1 better explains these points.

-Thoroughout the paper, there is a complete lack of distinction between tinnitus perception and tinnitus reaction. This is addressed indirectly in the questionnaire(s), but is hardly distinguished in the analysis, or in the discussion (except in line 333). Some statements are ambiguous, such as (line 332) "Stress [29,30], depression and anxiety [29,31,32] have been widely related to tinnitus". Which way do you mean the causation to occur? Does it affect your results?

Response: Thank you so much for this valuable comment. We have adjusted this part according to your suggestion, in order to make it more accurate.

- Statistics are probably not correctly used at times. If your duration of symptoms is "63.88 ± 94.14 months", it is a safe bet on my part that you are not using the right stats. For this (in multiple places), you might want to use median and interquartile range, it would be more informative to the reader too.

Response: Thank you very much for this comment. We have observed that some quantitative variables are not following normal distribution, so we have described them through medians and interquartile ranks (25%-75%). We included these changes in the material and method section and the new values have been recorded in our results.

Comments:

- The introduction was interesting and well written, but most of it was fairly irrelevant to the paper. If there is a need to reduce paper length, this would be a good place.

- What is the meaning of the exclusion "active otological illness". Since they have tinnitus, they seem to have an active illness. Needs precision.

- The requirement of being "caucasian" is a bit weird without justification. First of all, "White" should be used instead of "caucasian" I think, though I don't want to open that can of worms. More importantly, it limits the validity of the results to Whites only. This should be justified. It should be mentioned in the abstract. 

- The family history repeatedly refers to "three or less families" and I don't know what that means. Family members?

- How the groups of patients are referred to changes many times in the text. In just 3 paragraphs (around lines 240 to 280), the same (?) participants are referred to as "Outpatients", then as "offline participants" in the next paragraph, then as "hospital participants" in the next paragraph, then as outpatients again. Stick to one nomenclature throughout!

- Tinnitus is referred to as "disabling" multiple times, but I think you mean "debilitating".

- Line 321: "any otological diseases could eventually lead to tympanic membrane abnormalities". That seems to be a bit of an over statement?

Response: Thank you so much for all these valuable comments.

- We have adjusted the Introduction section, according to your suggestion

- We have included the concept “active otological illness” to refer to any acute condition affecting the ear, such as an active acute otitis media, chronic otitis or a vertigo crisis. In this regard, we have changed that term to “acute otological disease” and we have included a brief explanation of these in the text, to make it clearer.

- We have replaced Caucasians by “white Spanish population”, and included it in the abstract

- We have corrected this, as we were referring to “family members”. The word “families” has been changed to “relatives”.

- We have reviewed all the text and unify the way to refer to the outpatient cohort/survey.

- The word “disabling” has been replaced by “debilitating”

- This was a typo, since we meant that only some/few diseases could end up affecting the tympanic membrane.

Minor issues:

- line 117: why not include vertigo with dizziness?

- dyslipidemia (8 times) is spelled dyslipidaemia 3 times. Chose one!

Response: Thank you for your comment. After performing our first analysis, we did not include these variables, because we focused on psychoacoustic characteristics of tinnitus (part B of ESIT-SQ). However, we also think that both conditions are quite frequent and relevant in patients with chronic tinnitus. According to this, we have included information on the occurrence of vertigo in our results (tables 1, 3, 6)

In addition, we have unified the term “dyslipidemia”.

The paper has to be read by a native speaker. Here are a few of the points that made me stumble when reading. I don't normally lists stuff like this, but I would like to see this study published and be useful.

Line 30: However, these differences were partially confirmed

58  quality of life ant the annoyance (-> and)

58  the most extended questionnaires  (extensive?)

96  a specific call at social media

99  interfere the responses

100 addition disorder

134 high frecuency

135 prolfile

209 there were not differences

Table 2: worry on tinnitus: what does that mean?

Table 2 and other tables: "No answered" (-> no answer)

Table 2 and other tables: "Tinnitus increasing factors": what does that mean? Co-factors that will make an episode worse? I don't know.

Table 2: the line that states "Treatment" reports a value of 21.1/75.9 and again, I have no clue of the meaning of this.

296  others resulted inconsistent

375  it has been described a higher prevalence of anxiety

(then in the next line "there has been described")

393 there are not studies

Response: Thank you so much for taking the effort of listing all these typo. We have corrected them in the new version of the manuscript. We have also explained in the text the issues with Table 2, including the meaning of “tinnitus increasing factors”, and the ratio related to treatment (it has been expressed in percentage, and we have now included the “No answer” percentage, which was missing). In addition, we have updated all the terms according to the original ESIT-SQ version in English.

Reviewer 2 Report

Dear chief editor:

With special thanks, I read the manuscript entitled: "Age clustering may improve clinical profiling in Spanish patients with chronic tinnitus" that is submitted for the Journal of Clinical Medicine. Here, I have some viewpoints about the manuscript:

  • The title is intriguing; I think it must be changed and more relevant to the work.
  • I think comparing results of online surveys (for example in chronic tinnitus patients) with verbal or in-presence surveys is a critical, essential and necessary task; but I think this may be possible by a "cross-over" study design. At least here, I didn't see reaching the goal properly because of multiple biases, including selection bias: online survey group was more aware about their problem, they were younger than the other group and had more educational level, more time for completing the questionnaire, etc.
  • How the researchers did find the cases in two groups? Were the online participants with chronic tinnitus "truly" diagnosed by an otolaryngologist or audiologist? And also this problem in the other group: whether the outpatients cases did seek the otolaryngologist just because of their "chronic tinnitus" or other "otological problems" or even for other "ENT problems, such as obstructive sleep apnea"? All of these conditions would have confounding effects on the results!
  • What was the "condition" of the respondents in outpatient group? Did they have enough time, comfort, and more importantly "authority and liberty" for responding the questionnaire- comparable with the online group?
  • Why the researchers didn't matched the two groups according to main variables and confounding factors such as age, otological problems, educational level, and etc.?
  • Some of comorbidities, such as Meniere's disease (MD), and even sudden deafness (SSNHL) or hyperacusis, are very problematic diagnoses for the patients to be defined. I think such those comorbidities would not be possible for be investigated by a simple "Yes/No" question. What was the criteria of MD or SSNHL? Did they have pure tone audiometry? If yes, what was the mean of hearing loss, type of audiograms, and …
  • Also, what was "criteria" for anxiety or depression? These are not possible –or reliable- to be answered only by "Yes/No"!
  • Some grammatical or writing problems are there in the text that need to be corrected: for example in page 2, line 9.
  • What is the meaning of A level in demography tables?
  • In the discussion section, I think it is necessary comparing the results of other studies (with other questionnaires and also other diseases) that compared online vs. in-presence surveys. Many of studies that compared in the discussion section are not relevant to the present study. Albeit I think the major problem of the present study is its overall ambiguity.
  • In conclusion, I, as a readership or as a clinician or an otologist, cannot conclude that the online survey with those mentioned questionnaire is a good tool for "clinical profiling" in Spanish patients with chronic tinnitus with or without age clustering!

Thanks a lot

Author Response

  • The title is intriguing; I think it must be changed and more relevant to the work.

Response: Thank you for this value comment. We have adjusted the title according to our results.

  • I think comparing results of online surveys (for example in chronic tinnitus patients) with verbal or in-presence surveys is a critical, essential and necessary task; but I think this may be possible by a "cross-over" study design. At least here, I didn't see reaching the goal properly because of multiple biases, including selection bias: online survey group was more aware about their problem, they were younger than the other group and had more educational level, more time for completing the questionnaire, etc.

Response: Thank you for this valuable comment. We are aware of this concern too, as the use of new technologies can be a limiting factor, and could result in a selection bias. That was one of the reasons we decided to perform a cluster analysis, adjusting by age, to minimize this bias. The “cross-over” study desing is a very interesting point. However, in our study design this could not be posibble, since the online cohort consisted of volunteers that could not be tracked.

  • How the researchers did find the cases in two groups? Were the online participants with chronic tinnitus "truly" diagnosed by an otolaryngologist or audiologist? And also this problem in the other group: whether the outpatients cases did seek the otolaryngologist just because of their "chronic tinnitus" or other "otological problems" or even for other "ENT problems, such as obstructive sleep apnea"? All of these conditions would have confounding effects on the results!

Response: Individuals on the online survey were participant that self-reported tinnitus. A significant proportion of them were member of the Meniere Syndrome Spanish Association (ASMES) and they have been diagnosis with MD. Outpatients were individuals that reported tinnitus during the appointment at the hospital; regardless that chronic tinnitus was the primary complaint.

  • What was the "condition" of the respondents in outpatient group? Did they have enough time, comfort, and more importantly "authority and liberty" for responding the questionnaire- comparable with the online group?

Response: All the patients that were recruited in the hospital were offered the same conditions. All of them were asked to freely participate in this study, and were offered the chance to complete the questionnaire, once the medical visit was over, to have enough time to complete the survey. In addition, these patients could use a reserved area of the hospital, without the observation/influence of any clinician. Although these conditions could be slightly different to the online grup, we can assure that outpatients had time and privacy enough to complete the questionnaire, without external influences.

  • Why the researchers didn't matched the two groups according to main variables and confounding factors such as age, otological problems, educational level, and etc.?

Response: Thank you for this annotation. When we designed the study, we conceived and exploratory ecological approach, with the aim of comparing baseline from outpatients and compare them with the general population that report tinnitus. According to our preliminary results, we observed that age and Meniere’s disease (MD) were the main confounding factors, so we focused on them and decided to perform the cluster analysis, adjusting for age, and a stratified analysis for MD to solve both issues. Although there are some differences in terms of educational level, we did not performed any other statistical stratification because it would not offer more relevant information compared to the age-clustering.

  • Some of comorbidities, such as Meniere's disease (MD), and even sudden deafness (SSNHL) or hyperacusis, are very problematic diagnoses for the patients to be defined. I think such those comorbidities would not be possible for be investigated by a simple "Yes/No" question. What was the criteria of MD or SSNHL? Did they have pure tone audiometry? If yes, what was the mean of hearing loss, type of audiograms, and …

Response: Thank you for this comment. We are aware that the final diagnosis for MD, sudden SNHL or hyperacusis (and many other conditions recorded in the ESIT-SQ) should be supported by a clinician. Since the ESIT-SQ is a self-report questionaire, we understand that some patients could consider that they suffered from any of this problems. We also considered that these diseases (MD or SSNHL) are not as usually self-reported by patients, without a medical diagnosis compared with other conditions such as anxiety or depression. However, we consider that this results have to be analyzed cautiously, and help the observer to know the patient's feelings, rather than a definitive diagnosis of that disease. We have include this concern in the discussion section.

  • Also, what was "criteria" for anxiety or depression? These are not possible –or reliable- to be answered only by "Yes/No"!

Response: Thank you for this comment. We are aware that the final diagnosis for anxiety/depression (and the other conditions previously discused in this document) should be supported by a clinician. Since the ESIT-SQ is a self-report questionaire, we understand that some patients could consider that they suffered from any of this problems in some point. However, we also considered that these self-reported diseases have to be analyzed cautiously, and help the observer to know the patient's feelings, rather than a definitive diagnosis of that disease. We have include this concern in the discussion section.

  • Some grammatical or writing problems are there in the text that need to be corrected: for example in page 2, line 9.

Response: Thank you for this comment. We have reviewed the whole manuscript and corrected all the grammar and vocabulary mystakes.

  • What is the meaning of A level in demography tables?

Reponse: This was a typo, since we refer to “High school”. This has been corrected in the manuscript, according to the original ESIT-SQ version in English.

  • In the discussion section, I think it is necessary comparing the results of other studies (with other questionnaires and also other diseases) that compared online vs. in-presence surveys. Many of studies that compared in the discussion section are not relevant to the present study. Albeit I think the major problem of the present study is its overall ambiguity.

Response: Thanks you for this comment. Regrettably, there are no researches that have been carry out using ESIT-SQ. However, other questionnaires have been used to study the characteristics of tinnitus. We have added the in the discussion part.

  • In conclusion, I, as a readership or as a clinician or an otologist, cannot conclude that the online survey with those mentioned questionnaire is a good tool for "clinical profiling" in Spanish patients with chronic tinnitus with or without age clustering!

Response: We entirely agree with you. We have included this in the conclusions of the study as we state in the abstract section: “Self-reported tinnitus surveys are a low confidence source for tinnitus phenotyping. Additional clinical evaluation is needed for tinnitus research to reach the diagnosis”

Round 2

Reviewer 2 Report

Thanks. The changes improved the quality of the manuscript, but I think it is too long. I strongly suggest re-writing  and more summarizing it.

thanks again 

Author Response

Thanks. The changes improved the quality of the manuscript, but I think it is too long. I strongly suggest re-writing and more summarizing it.

Response: Thank you for taking the time to review our article. According to your suggestions, we have re-write and summarized the manuscript, focusing in the discussion section.